# A PCR primer design method for identifying spider mite species using *k*-mer counting

**Tomoko Matsuda**[1]*, **Hironori Sakamoto**[2], **Takumi Kayukawa**[3], **Yasuki Kitashima**[4], **Toshinori Kozaki**[4], **Tetsuo Gotoh**[4,5]

**1** Research and Development Department, Nihon BioData Corporation, Kawasaki, Kanagawa, Japan, **2** Biodiversity Division, National Institute for Environmental Studies, Tsukuba, Ibaraki, Japan, **3** Institute of Agrobiological Sciences, National Agriculture and Food Research Organization, Tsukuba, Ibaraki, Japan, **4** Faculty of Agriculture, Ibaraki University, Ami, Ibaraki, Japan, **5** Faculty of Economics, Ryutsu Keizai University, Ryugasaki, Ibaraki, Japan

* tomokom@nbiodata.com

## Abstract

Using PCR to distinguish closely related species can be difficult because they may have very similar genomes. Advances in bioinformatics make it possible to design PCR primers that are species-specific. In this study, we developed a bioinformatics method for extracting species-specific primer candidate sequences (i.e., unpaired primers that were specific to a single species) from RNA-Seq data sets of 19 species of spider mites (Acari, Tetranychidae). Using *k*-mer counting, we obtained between 257 and 48,621 species-specific unpaired primer candidates for the 19 species. We then manually obtained a second primer that was also species-specific. The primer pairs were then confirmed to work in the target species and not to work in the non-target species. Finally, species-specific primer pairs were obtained for 17 of the 19 species tested. Such species-specific primers may be used for practical species discrimination by optimizing multiplex PCR. Our primer design method is expected to be applicable to other taxa.

## Introduction

Polymerase chain reaction (PCR) is a method for amplifying DNA fragments *in vitro* [1]. Because of its convenience and low cost, PCR has become a standard technique not only for biomedical, infectious disease, and forensic applications but also in other fields of biology, such as agricultural and ecological research. Among its many uses, PCR can be used to distinguish morphologically similar species. For example, PCR-RFLP (restriction fragment length polymorphism) distinguishes closely related species by using a single restriction enzyme to produce fragments of different lengths from DNA markers [2–4]. Another method is to design species-specific primers that can only be amplified in certain species and to identify the species based on successful PCR amplification and correct product size [5–7]. In that case, designing PCR primers

**Data availability statement:** The RNA-Seq datasets DRA007145 and DRA018635 are publicly available at the following links: DRA007145: https://www.ncbi.nlm.nih.gov/sra/?term=DRA007145 DRA018635: https://www.ncbi.nlm.nih.gov/sra/?term=DRA018635. The sequence data related to Table 2 are also publicly available via the accession numbers listed in Table 2.

**Funding:** This study was supported in part by the Japan Society for the Promotion of Science (JSPS) KAKENHI Grant Number JP25292033 (TG) and JP17H03775 (TG). JSPS website: https://www.jsps.go.jp/english/. The funders had no role in study design, data collection and analysis, decision to publish, or preparation of the manuscript.

**Competing interests:** Tomoko Matsuda is a member of a commercial company: Nihon BioData Corporation. This does not alter the authors' adherence to PLOS ONE policies on sharing data and materials. The authors have also declared that no competing interests exist.

to be specific to the target is essential. Several web-based services and stand-alone software programs, such as Primer-BLAST [8], PrimerSNP [9], MultiMPrimer3 [10], and ssGeneFinder [11], are designed to optimize primer design for PCR. Furthermore, advances in bioinformatics make it possible to obtain more efficient primer candidates and reduce the amount of false-positive amplification [12–14].

One of the methods for designing primers from large sequence data uses $k$-mer counting. A $k$-mer is an oligonucleotide of length $k$ and is utilized for various bioinformatics analyses, such as de Bruijn graph assembly [15–17], estimation of genome size [18–20], and estimation of species abundance in metagenomic samples [21–23]. Some studies have provided ways to design primers or probes to detect strains or species using $k$-mers from large-scale sequence data. PriMux software predicts degenerate primers and probes from a large target set with very low homology and poor alignment (full-length genomes of all serotypes of the dengue virus, 2,863 sequences) [24]. The method allows the selection of primers optimized for viral genomes of different serotypes. It does not require multiple sequence alignment, which involves computation time and memory space, and instead uses $k$-mer analysis. PathogenMIPer software is a program for designing unique and specific molecular inversion probe (MIP) oligonucleotides for pathogen identification and detection [25]. Probes generated by PathogenMIPer were able to detect 24 human papillomavirus (HPV) types in clinical samples with 100% sensitivity and no false positives. RUCS is a program that identifies PCR primer pairs and probes present in one genomic dataset but not another [26]. These studies have targeted viruses and pathogens with relatively small genomes. For eukaryotes, which have larger genomes than prokaryotes, more time is generally required for $k$-mer analyses. In addition, in the case of eukaryotes, whole genome sequencing is often performed only for common species. For example, in the case of the spider mite family Tetranychidae, only four assembled genomes (*Tetranychus urticae* Koch [27], *Tetranychus cinnabarinus* (Boisduval) [28], *Panonychus citri* (McGregor), *Tetranychus truncatus* Ehara) are available in the Sequence Read Archive (SRA). Using whole genome data would likely increase the number of available targets and the number of sequences and non-transcribed regions considered. However, publicly available sequencing resources for spider mites (Acari, Tetranychidae) are limited, with only 13 species having whole genome short-read data deposited in the SRA, compared to 79 species with available RNA-Seq data [29] (according to the NCBI SRA database accessed on 24 December 2024). In a previous study, we performed RNA-Seq analysis on 72 species (73 strains) of spider mites. The resulting data have been made publicly available in DRA/SRA/ERA databases under the accession number DRA007145. Once a method for designing species-specific primers utilizing these RNA-Seq data is established, it could be applied to other taxa.

The purpose of the present study was to develop bioinformatics methods for extracting species-specific primer candidate sequences from the whole-transcriptome RNA-Seq data of spider mites. Spider mites of the family Tetranychidae include some agricultural pest species that cause severe economic losses worldwide [30,31]. Accurate species identification is instrumental for effective pest control, but closely related

mite species are difficult to distinguish by morphological characters. In particular, some species of the genus *Tetranychus* are morphologically similar, differing only in the diameter of the aedeagal knob in males [32]. However, most specimens collected in the field are adult females because of a female-biased sex ratio [33]. To overcome this problem, several molecular methods have been used to distinguish spider mite species [34], such as DNA barcoding [35–40], PCR-RFLP [41,42], multiplex PCR [43], and real-time PCR [44]. A limitation with most of these techniques is that they do not work well for discriminating species across more than two genera [34]. In the present study, we developed a method for discriminating species in five genera in the family Tetranychidae. First, species-specific primer candidates were extracted from RNA-Seq assemblies using *k*-mer counting. We then manually obtained a second primer that was also species-specific (Fig 1). To validate the primers, we performed PCR amplification and confirmed the amplification in the target species and the absence of amplification in non-target species. Finally, species-specific primers were obtained for 17 of the 19 species tested. These species-specific primers may be used for species discrimination by optimizing multiplex PCR.

## Materials and methods

### Spider mites

Nineteen species of spider mites were used in this study (Table 1). The green and red forms of *T. urticae* were treated as separate species in this study. Mite samples were maintained on leaf discs of common bean (*Phaseolus vulgaris* L.) or the original host plants placed on a water-saturated polyurethane mat in a plastic dish (90 mm diameter, 20 mm depth) at 25°C under a 16L-8D photoperiod until analysis. Voucher specimens were prepared as described previously [45] and were preserved in the Laboratory of Applied Entomology and Zoology, Faculty of Agriculture, Ibaraki University.

### Identification of species-specific primers

The protocol for identifying the species-specific primers and PCR validation used in this study has been deposited and published on protocols.io with DOI: dx.doi.org/10.17504/protocols.io.q26g7m78qgwz/v1.

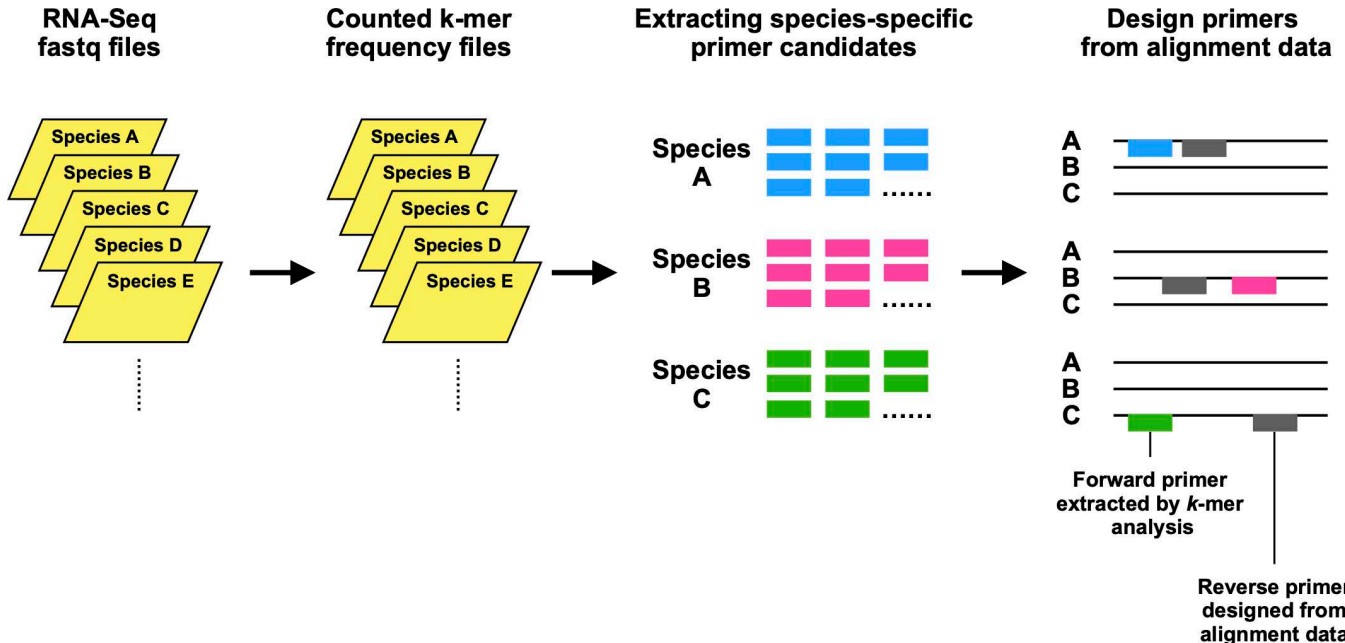

**Fig 1. Primer design methods for the detection of spider mite species using *k*-mer counting.**

**Table 1. Classification and sources of tetranychid mites used in this study.**

| Genus | Species | Collection date | Locality | Host plant | Voucher specimen no.[a] | Number of species-specific primer candidates |
|---|---|---|---|---|---|---|
| *Panonychus* | *P. citri* (McGregor) | May 6, 1993 | Ibaraki, Japan | *Ilex crenata* | 226 | 1,334 |
| | *P. mori* Yokoyama | Apr. 22, 2007 | Hokkaido, Japan | *Morus australis* | 239 | 3,333 |
| | *P. ulmi* (Koch) | Aug. 14, 2020 | Iwate, Japan | *Malus pumila* | 938 | 1,842 |
| | *P. osmanthi* Ehara & Gotoh | June 15, 2010 | Tokyo, Japan | *Osmanthus fragrans* | 600 | 257 |
| *Schizotetranychus* | *S. shii* (Ehara) | Aug. 16, 2020 | Ibaraki, Japan | *Castanopsis sieboldii* | 940 | 32,623 |
| *Eotetranychus* | *E. nomurai* Ehara | Aug. 27, 2020 | Ibaraki, Japan | *Celtis sinensis* | 932 | 48,621 |
| | *E. celtis* Ehara | Aug. 27, 2020 | Ibaraki, Japan | *Aphananthe aspera* | 931 | 38,701 |
| *Oligonychus* | *O. castaneae* Ehara & Gotoh | Oct. 5, 2017 | Ibaraki, Japan | *Castanea crenata* | 878 | 2,143 |
| | *O. ilicis* (McGregor) | Oct. 10, 2017 | Ibaraki, Japan | *Rhododendron* sp. | 935 | 5,025 |
| | *O. coffeae* (Nietner) | May 30, 2005 | Okinawa, Japan | *Mangifera indica* | 78 | 7,839 |
| | *O. gotohi* Ehara | July 30, 2000 | Chiba, Japan | *Lithocarpus edulis* | 96 | 2,484 |
| | *O. amiensis* Ehara & Gotoh | July 13, 2005 | Ibaraki, Japan | *Lithocarpus edulis* | 116 | 4,109 |
| *Tetranychus* | *T. kanzawai* Kishida | May 19, 1993 | Shizuoka, Japan | *Thea sinensis* | 158 | 1,089 |
| | *T. parakanzawai* Ehara | Aug. 16, 2009 | Chiba, Japan | *Morus australis* | 339 | 400 |
| | *T. urticae* Koch (red form) | Aug. 27, 2001 | Nagano, Japan | *Dianthus* sp. | 171 | 505 |
| | *T. urticae* Koch (green form) | July 16, 2001 | Hokkaido, Japan | *Citrullus lanatus* | 181 | 437 |
| | *T. truncatus* Ehara | May 8, 2004 | Kyoto, Japan | *Solanum nigrum* | 195 | 2,950 |
| | *T. pueraricola* Ehara & Gotoh | Oct. 23, 1993 | Ibaraki, Japan | *Pueraria montana* | 203 | 456 |
| | *T. piercei* McGregor | Dec. 20, 2007 | Okinawa, Japan | *Cucumis melo* | 14 | 6,840 |

[a]Voucher specimens are preserved at the Laboratory of Applied Entomology and Zoology, Faculty of Agriculture, Ibaraki University, under a serial number.

**RNA-Sequencing.** RNA-Seq dataset for 16 of the 19 species was reused from our previous work. The BioSample accession number of each RNA-Seq data is associated with the DRA accession number: DRA007145 (https://www.ncbi.nlm.nih.gov/sra/?term=DRA007145, S1 Table). RNA-Seq data for the remaining three species (*Eotetranychus nomurai* Ehara, *Eotetranychus celtis* Ehara, *Oligonychus castaneae* Ehara & Gotoh) were newly acquired. Total RNA was extracted from whole bodies of 100–200 adult females of the same population using an RNeasy Micro Kit (Qiagen, Valencia, CA, USA). Live female individuals for RNA samples and female individuals for voucher specimens were obtained from the same leaf discs and plants. The quantity and quality of the total RNA were evaluated using Agilent RNA ScreenTape System (Agilent Technologies, Santa Clara, CA, USA). The cDNA libraries were prepared from the total RNA using the TruSeq RNA sample prep kit (Illumina, San Diego, CA, USA), and the single ends were sequenced for 75 cycles on the NextSeq 500 sequencing platform (Illumina). All the reads were deposited in the DDBJ Sequence Read Archive. The BioSample accession number of each RNA-Seq dataset is associated with the DRA accession number: DRA018635 (https://www.ncbi.nlm.nih.gov/sra/?term=DRA018635, S1 Table).

***de novo* assembly.** The workflow of the subsequent bioinformatics analysis is shown in Fig 2. The sequence reads were trimmed by fastx_trimmer of the FASTX-Toolkit [46] (parameter: -f 15) and by fastq_quality_trimmer (parameters: -t 28 and -l 40), and then filtered by fastq_quality_filter (parameters -q 28 and -p 80). The processed sequence reads were assembled into contigs per species by Bridger [47] with the following command: Bridger --seqType fq --output [output_dir] --single [reads.fq] --CPU 16. Contigs with 95% or more similarity were judged as redundant and removed by CD-HIT [48]. The open reading frames (ORFs) were identified by TransDecoder [49].

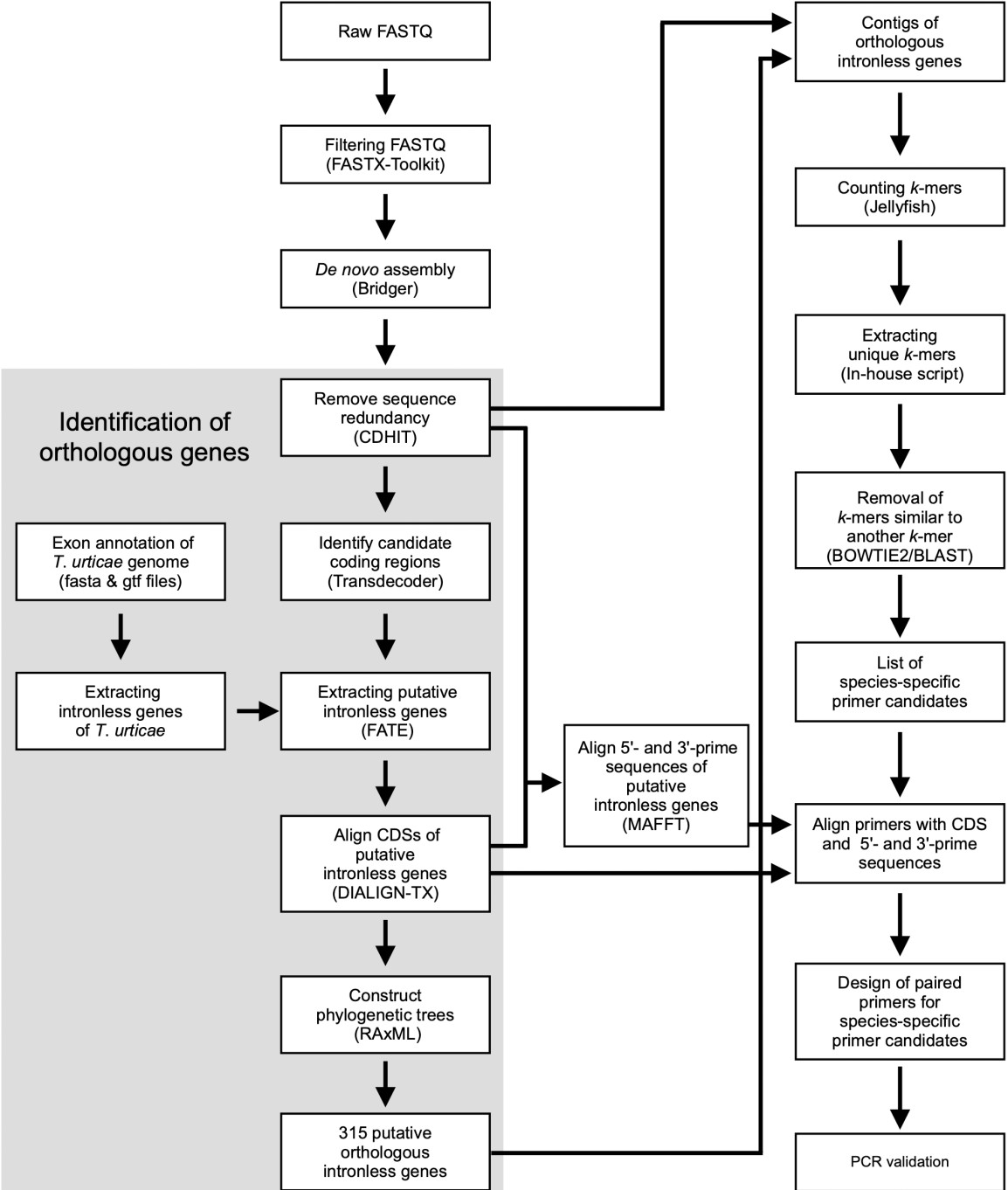

**Fig 2. Workflow diagram for extracting species-specific primers from RNA-Seq data using *k*-mer counting.**

**Identification of putative orthologous genes.** A problem with using RNA-Seq data for primer design is that they don't include introns, which are present in genomic DNA. Introns in the primer sequence itself or between the primers could prevent amplification. Therefore, we extracted species-specific primers from intronless genes, which were identified from the exon annotations of the *T. urticae* genome [27]. The ORFs of the 19 species were annotated by FATE [50] with

the TBLASTN engine and other default parameters against the intronless genes of the *T. urticae* genome [27]. A total of 431 intronless genes were annotated, and orthologous ORFs were aligned by DIALIGN-TX [51] with the -L option. Despite recent advances, existing ortholog detection methods still suffer from false-positive and false-negative rates. We therefore constructed a phylogenetic tree using RAxML [52] to differentiate between orthologous and paralogous genes. A gene was classified as a paralog if its phylogenetic relationship distinctly deviated from the established relationships among known spider mite species in the family Tetranychidae [29]. One hundred and sixteen genes that appeared to be paralogs were identified visually. After removing paralogous genes, 315 putative orthologous intronless genes were used for *k*-mer analysis.

**_k_-mer analysis.** Contigs clustered with CD-HIT, which were assigned to orthologous intronless genes (315 genes), were used for the *k*-mer analysis. *K*-mer analysis with *k* set to 30 was performed by Jellyfish [53] for each species. The command in Jellyfish is: jellyfish count -m 30 -s 10000 -t 4 -o [output_file] [input_fasta].

**Extracting unique _k_-mers.** *K*-mers obtained from Jellyfish were filtered using the following requirements. Primer specificity for species-specific detection:

1. If a *k*-mer matched another *k*-mer within the same species or a *k*-mer of another species, the *k*-mer was removed.

   Tm (primer melting temperature):

2. If the Tm value of a *k*-mer was less than 67 or greater than 82, the *k*-mer was removed.

   GC Content: Preferably in the range of 40%–60%. Include 1–2 G or C bases at the 3' end to improve binding stability; however, avoid excessive GC content at the 3' end to prevent non-specific binding.

3. If the GC content of a full-length *k*-mer (30-mers) was less than 11 or greater than 17, the *k*-mer was removed.

4. If the GC content of the former or latter half of a *k*-mer (15-mers) was less than four or greater than 10, the *k*-mer was removed.

5. If both the 5' and 3' ends of a *k*-mer were A or T, the *k*-mer was removed.

6. If both the 5' and 3' ends of a *k*-mer contained three or more of G or C, the *k*-mer was removed.

7. If both the 5' and 3' ends of a *k*-mer contained three or more of A or T, the *k*-mer was removed.

   Avoid Repeats and Runs: Prevent intra-primer and inter-primer complementarity to reduce primer-dimer formation.

8. If a *k*-mer contained four or more runs of a single base (e.g., AAAAA or GGGG), the *k*-mer was removed.

9. If a *k*-mer contained more than three dinucleotide repeats (e.g., ATATAT or CGCGCG), the *k*-mer was removed.

**Filtering _k_-mers.** The filtering of unique *k*-mers, which serve as candidates for species-specific primers, was performed using both Bowtie2 [54] and BLASTN [55]. Bowtie2 offers a significant advantage in computational speed, allowing for rapid initial filtering. In contrast, BLASTN permits more mismatches and supports finer matching criteria, providing greater flexibility and accuracy in sequence comparison. By first applying Bowtie2 for preliminary filtering and then refining the reduced set of *k*-mers with BLASTN, we optimize both computational performance and precision in the filtering process. Extracted unique *k*-mer sets of 19 species were mapped to contigs of the 19 assemblies using Bowtie2 software. When a *k*-mer was mapped multiple times to contigs of the same species or matched 26-mers or more to other species, the *k*-mer was removed. Then, BLASTN (blastn-short) searches were performed between the *k*-mers filtered by Bowtie2 (query) and contigs of the 19 assemblies after clustering with CD-HIT (database). When a *k*-mer aligned multiple times or 24–29 bp to a contig of the same species, the *k*-mer was removed. Furthermore, *k*-mers that matched contigs of other species by more than 25 bp were removed. The *k*-mers remaining after filtering by Bowtie2 and BLASTN were considered species-specific primer candidates.

## PCR validation

**Finding potential primer pairs from alignment.** Computationally identified species-specific *k*-mers occasionally exhibited unintended sequence matches to other species during sequence alignment checks, reducing their specificity. For these reasons, PCR primers to be paired with species-specific primer candidates were designed from the alignment of the coding sequences (CDSs) and the 5' and 3' UTR sequences (Table 2, S2 Table, S1 File). CDSs were aligned using DIALIGN-TX, which considers amino acid sequences to improve alignment accuracy, as described in the section 'Identification of putative orthologous genes.' In contrast, 3' and 5' UTR sequences were aligned using MAFFT [56], a widely used alignment tool, as considering amino acid sequences is not necessary for these non-coding regions. Selection criteria included maintaining an appropriate distance for amplification and ensuring primer specificity within the transcriptome to minimize cross-amplification. This approach enabled the practical refinement of primer design beyond automated *k*-mer pairing. In some cases, the position and length of the species-specific primers were modified according to the alignment. To confirm the uniqueness of the species-specific primers, we conducted additional BLASTN searches to compare the species-specific primers with the contigs from the 19 assemblies after clustering with CD-HIT.

**DNA extraction and PCR amplification.** Total DNA was extracted from the whole bodies of individual female spider mites using a QIAamp DNA Micro Kit (Qiagen, Valencia, CA, USA). To ensure the quality of the extracted DNA and to provide a positive control for the PCR reactions, we tested the DNA using primer sets known to amplify spider mite DNA. Specifically, we targeted the cytochrome c oxidase subunit I (COI) gene of mitochondrial DNA and the 28S nuclear ribosomal RNA (rRNA) gene [45]. The successful amplification of these markers confirmed that the extracted DNA is of sufficient quality for reliable PCR analysis. The PCR primers are shown in Table 2 and S2 Table. A pair of primers, designed to amplify a single species, was tested against all 19 species used in this study. When a gel band was observed for only a single species, the primer set was considered to be effective in detecting the species. PCR reactions were performed in a 20 μL mixture containing 1 μL of DNA template, 0.4 μL of KOD FX Neo (1 U/μL, Toyobo, Osaka, Japan), 0.6 μL of each primer (10 pmol/μL each), 4 μL of 2 mM dNTPs (Toyobo), 10 μL of 2 × PCR Buffer for KOD FX Neo (Toyobo), and 3.4 μL of distilled water. The PCR cycling parameters were as follows: 2 min of denaturation at 94°C, 35 cycles of 10 sec at 98°C, and 1 min at 68°C. An additional 1 min at 68°C was allowed for final strand elongation. PCR products were visualized by electrophoresis on an agarose gel. A 100-bp DNA ladder (Takara Bio, Shiga, Japan) was used as a molecular size marker.

**Sequencing.** The PCR product was sequenced only when the target species was successfully amplified. PCR products were purified using Sephacryl S-300 High Resolution (GE Healthcare, Chicago, IL, USA) and directly sequenced. Sequencing was carried out in both directions using the amplifying primers with the BigDye Terminator Cycle Sequencing Kit v.3.1 (Applied Biosystems, Foster City, CA) and on an ABI 3130xl Genetic Analyzer (Applied Biosystems). Then, BLASTN searches were performed between the sequences of the PCR products (query) and the contigs that contain species-specific primers (database) to confirm sequence similarity. The sequences of the PCR products have been deposited in the DDBJ/EMBL/GenBank International Nucleotide Sequence Databases under accession numbers LC867280 to LC867298.

## Results

RNA-Seq data were obtained using RNA extracted from 100–200 adult females of each spider mite species. Pooling individuals can introduce intraspecific variation, which could potentially affect the accuracy of downstream analyses. However, as most RNA-Seq data are derived from transcribed regions, they are less affected by noise from intraspecific variation compared to whole-genome DNA-Seq data, which include non-coding or non-transcribed regions. This makes RNA-Seq data particularly advantageous for detecting species-specific markers. The *de novo* assembly of the RNA-Seq data used in this study yielded an N50 length of 1,369–2,256 bp and a maximum contig length of 11,286–24,030 bp. These values are comparable to those reported in other spider mite studies [29,57]. Based on this level of assembly quality, we

**Table 2. Species-specific primer sequences and results of PCR amplification.**

| Species name | Forward primer name | Forward primer sequence[a] | Reverse primer name | Reverse primer sequence[a] | Expected PCR product size | PCR amplification success | Species PCR-amplified | NCBI Reference Sequence | Accession numbers of PCR product sequences | Percentage of identical matches (%) | Alignment length (bp) |
|---|---|---|---|---|---|---|---|---|---|---|---|
| P. citri | Pci6_F1 | ACGATGGAATGACCTGTAGTGTAAAGTTCGC | Pci6_R1 | GCGCGGTGTCGGTGTTTACGTTACTGG | 604 | No | P. citri, P. osmanthi | NW_015449934.1 | – | – | – |
| P. mori | Pmo1_F2 | ATGGAACTTGTGATGTACCATATTCGCAGAC | Pmo1_R2 | AACTCATTGACCGCCTCCTGATCAATTCC | 102 | Yes | P. mori | XM_025162220.1 | LC867280 | 98.82 | 85 |
| P. ulmi | Pul7_F1 | TCATATTTCCAAATACTCTGCTCAAAGTTCACC | Pul7_R1 | AAGTTGCTTTAGGCGTTTGTGGTTTGAAGG | 725 | Yes | P. ulmi | XM_015925916.2 | LC867281 | 99.71 | 692 |
| P. osmanthi | Pos5_F1 | TGTTTTGTCTTTACCGCTTAGAGTGAAGAAGG | Pos5_R1 | TTTCCTCTTAAGAGTAGCCGGAACAATTTCAGC | 504 | Yes | P. osmanthi | XM_015932902.2 | LC867282 | 100 | 505 |
| S. shii | Ssh7_F1 | AGTGAAGAAGGGTTTAAACATCGCCAATAAACTC | Ssh7_R1 | AGTTACTTCACCCCACATTCGGGAACC | 702 | Yes | S. shii | XM_015937235.2 | LC867283 | 100 | 664 |
| E. nomurai | Eno6_F1 | CGTTCGGCTCAATTCACTGAACCTGAG | Eno6_R1 | AACTGAAGATGCGTATGTGAGGACGGTC | 600 | Yes | E. nomurai | XM_015939562.2 | LC867284 | 99.81 | 524 |
| E. celtis | Ece3_F1 | AGAATTGGAAAATGTTCAAACGAAGATGAAATTGGTTC | Ece3_R1 | AGCTTGATGTAGTTGGGCAAGCTCAGC | 300 | Yes | E. celtis | XM_015937235.2 | LC867285 | 99.67 | 301 |
|  | Ece5_F2 | TGGCCATTCTTGAATATGGACAGAATCAAGAC | Ece5_R2 | TCGATTGATCCCGAGATTGGACAGGG | 503 | Yes | E. celtis | XM_015937235.2 | LC867286 | 99.61 | 506 |
| O. castaneae | Oca4_F1 | GAGAAGGATCGTATGGCATAACCATCTCG | Oca4_R1 | ACCATGTATGGACAAACCAATTGCTGGTTGAG | 399 | Yes | O. castaneae | XM_015940454.2 | LC867287 | 98.87 | 355 |
|  | Oca1_F1 | TCATCAAGGTGCTGCTGATTCATTCACAACTTC | Oca1_R1 | TGATCACTGATTCTTTACGAGTTTTTCAACACACAC | 109 | Yes | O. castaneae | XM_015930636.2 | LC867288 | 100 | 13 |
| O. ilicis | Oil2_F1 | ACATATAGAAAGCTTGTTAATGACAGTCCGTCG | Oil2_R1 | AGTTGTAATTGTATTCCTATGGTAATTAGCGTGATG | 205 | Yes | O. ilicis | XM_015937235.2 | LC867289 | 99.52 | 206 |
| O. coffeae | Oco1_F1 | CCAGATCGCAGATTCAGACTCTGTTATACG | Oco1_R1 | TGAAGGTGATGTAATTGCTTAATGTTTATAATTATTCCTC | 109 | Yes | O. coffeae | XM_015937235.2 | LC867290 | 82.9 | 76 |
| O. gotohi | Ogo7_F1 | TCAGAGGGTTGTTCCAGAAAGTGAATTACCG | Ogo7_R1 | AGAAATTCTAATCCGGTTACAAAGATATTATGAGCG | 703 | Yes | O. gotohi | XM_015928528.2 | LC867291 | 99.85 | 676 |
| O. amiensis | Oam2_F1 | ACTTAGTCTTTTGAATTTTCTCAACTGTAAACTAACTG | Oam2_R1 | TCCGATCTTTCAAGGTAATGCCAATGACG | 199 | Yes | O. amiensis | XM_015937235.2 | LC867292 | 99.5 | 200 |
| T. kanzawai | Tka8_F1 | TAATTGTACCTAATATTGTTTATGAGGATGCTATTAGC | Tka8_R1 | CATCTTGTAAACAAATGTCGATTTGTATCGATTAACTC | 786 | Yes | T. kanzawai | XM_015937235.2 | LC867293 | 99.86 | 723 |
| T. parakanzawai | Tpar2_F1 | GTCACAAATTTTGCGAAGCAGTGAAATCACTG | Tpar2_R1 | ATCCACTGATTCGACTGGCGCAAGTC | 203 | Yes | T. parakanzawai | XM_015939595.2 | LC867294 | 87.82 | 156 |

**Table 2.** (Continued)

| Species name | Forward primer name | Forward primer sequence[a] | Reverse primer name | Reverse primer sequence[a] | Expected PCR product size | PCR amplification success | Species PCR-amplified | NCBI Reference Sequence | Accession numbers of PCR product sequences | Results of BLASTN search[b] | |
|---|---|---|---|---|---|---|---|---|---|---|---|
| | | | | | | | | | | Percentage of identical matches (%) | Alignment length (bp) |
| *T. urticae* (red form) | TurR6_F1 | **CAGTCGACAGAATTGGC-GAAGTCCAAG** | TurR6_R1 | TCTCTTCAGGTAAAC-CAATAATTGAAATTC-CAGC | 612 | Yes | *T. urticae* (red form) | NW_015449954.1 | LC867295 | 96.52 | 604 |
| *T. urticae* (green form) | TurG5_F2 | TCATCAGGAAAAGA-CAAAGAATATGAAAATAT-CATTAGG | TurG5_R2 | **ACAAGTCTGGGTT-GAGATGGAGCTTCC** | 511 | No | Non-amplified | XM_015926370.2 | – | – | – |
| *T. truncatus* | Ttr4_F1 | **TGATGAAGATTCGG-CAACTCGACGTTTCAG** | Ttr4_R1 | CTCCGAGGCAACCT-GAAGGTTTTGG | 415 | Yes | *T. truncatus* | XM_0159281 25.2 | LC867296 | 100 | 223 |
| *T. puer-aricola* | Tpu6_F1 | TGTCAACATATGATCCT-GAGAAGCAAGATGG | Tpu6_R1 | **ACAAGCG-CATTTATCTAGC-GTTCATGCAGC** | 605 | Yes | *T. pueraricola* | XM_015940290.2 | LC867297 | 99.66 | 593 |
| *T. piercei* | Tpi1_F1 | **TCACCTTTTTTTGGTAT-GTTTAGGAAGCACATC** | Tpi1_R1 | ATACTGATATCAAA-CATTTTGAGGTAAGTG-ACAAGATTG | 99 | Yes | *T. piercei* | XM_015937235.2 | LC867298 | 80.88 | 68 |

[a] All primer sequences used in this study are listed in S2 Table, and gel images are provided in S3 File. Primer candidates designed based on the results of *k*-mer analysis are indicated in bold font.

[b] Results of BLASTN searches between PCR-amplified products and NCBI reference sequences.

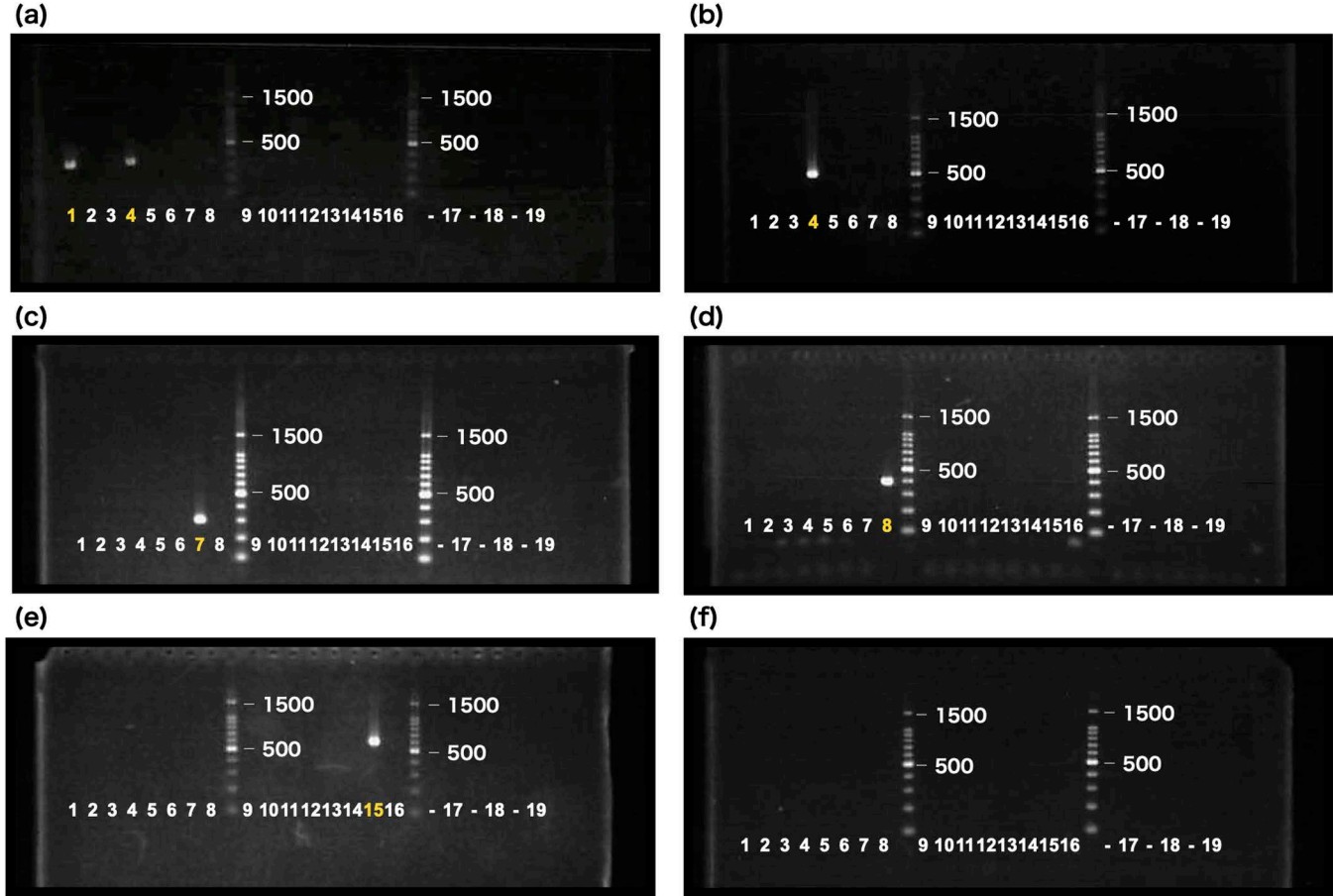

**Fig 3. Agarose gel showing the results of PCR amplification using species-specific primers.** Lanes 1: *P. citri*, 2: *P. mori*, 3: *P. ulmi*, 4: *P. osmanthi*, 5: *S. shii*, 6: *E. nomurai*, 7: *E. celtis*, 8: *O. castaneae*, 9: *O. ilicis*, 10: *O. coffeae*, 11: *O. gotohi*, 12: *O. amiensis*, 13: *T. kanzawai*, 14: *T. parakanzawai*, 15: *T. urticae* (red-form), 16: *T. urticae* (green-form), 17: *T. truncatus*, 18: *T. pueraricola*, 19: *T. piercei*. (a) Bands of *P. citri* (lane 1) and *P. osmanthi* (lane 4) were amplified with primers Pci6_F1 and Pci6_R1. (b) The band of *P. osmanthi* (lane 4) was amplified with primers Pos5_F1 and Pos5_R1. (c) The band of *E. celtis* (lane 7) was amplified with primers Ece3_F1 and Ece3_R1. (d) The band of *O. castaneae* (lane 8) was amplified with primers Oca4_F1 and Oca4_R1. (e) The band of *T. urticae* (red form, lane 15) was amplified with primers TurR6_F1 and TurR6_R1. (f) Primers TurG5_F2 and TurG5_R2 did not amplify any products. Additional results are provided in the supplementary data (S3 File).

determined that the data were appropriate for robust *k*-mer analysis. *K*-mer analysis (Fig 2) provided between 257 and 48,621 species-specific primer candidates for the 19 species (Table 1, S2 Table, and S2 File). Primer pairs were designed by manually examining alignment files to locate positions corresponding to species-specific sequences. Alignments were reviewed to confirm the specificity of candidate primers across the tested species. The primer pairs were initially designed randomly. However, since sequence alignments were carefully examined during the primer design process, additional primers were occasionally designed from the same contig when a particular alignment was deemed highly suitable for primer design. As a result, multiple primers were designed from the same transcript in certain cases, such as XM_015937235.2 and NW_015449938.1, as shown in Table 2 and S2 Table. Finally, 43 PCR primers were designed to target species-specific sequences.

The species specificity of the primers was confirmed by amplification in the target species and the absence of amplification in non-target species. PCR validation showed that 19 of the 43 primer pairs amplified only the target species (Fig 3, Table 2, S2 Table, S3 File, and S1 Raw Images). The remaining 26 primer pairs amplified non-specific products or

did not amplify anything. For most species, a species-specific primer pair was successfully developed after multiple cycles of design and testing. However, despite multiple attempts, the specific primer pairs designed for *P. citri* and the green form of *T. urticae* failed to amplify the target species specifically. Sequence similarity between the sequences of PCR products and the contigs that contain species-specific primers was greater than 96% in 17 of the 19 cases (Table 2, S2 Table).

## Discussion

Our *k*-mer analysis generated between hundreds and tens of thousands of species-specific primer candidates for each of the 19 species tested (Table 1). As *k*-mer counting is used for alignment-free phylogenetic inference [58–60], *k*-mer frequency profiles tend to be similar among closely related species. As expected, for closely related species, the number of primer candidates was at the low end of the range. This was the case for 1) *T. pueraricola* and the green and red forms of *T. urticae*, 2) *Tetranychus kanzawai* Kishida and *T. parakanzawai* Ehara, and 3) *Panonychus citri* (McGregor) and *P. osmanthi* Ehara & Gotoh. Nineteen of the 43 primer pairs amplified only the target species, whereas the remaining 26 primer pairs either amplified non-specific products or did not amplify anything. Finally, species-specific primers were obtained for all but two of the 19 species tested: *P. citri* and the green form of *T. urticae* (Tables 2 and S2).

Primers unique to the RNA-Seq data may also match sequences in non-transcribed regions of the genome, and therefore, may result in non-targeted or non-specific amplification when genomic DNA is used as the template. For example, in this study, primers specific to the green form of *T. urticae* amplified products in *Tetranychus* species other than the green form of *T. urticae*, and primers specific for *P. mori* amplified a product in *T. kanzawai* (S2 Table). We conducted *k*-mer analyses with 30 bp *k*-mers (i.e., 30 bp primers), which are longer than the commonly used length (18–24 bp) and therefore provide greater specificity [61]. However, species-specific amplification was not successful for *P. citri* and the green form of *T. urticae*. This may be due to the close relationship between the green form of *T. urticae*, *T. pueraricola*, and the red form of *T. urticae*, as well as between *P. citri* and *P. osmanthi*. In contrast, species-specific primers were successfully designed and experimentally validated for *T. kanzawai* and its close relative *T. parakanzawai*, as well as for *O. gotohi* and *O. amiensis*. PCR results for closely related species not tested in this study could differ from the expected outcomes. This limitation is not unique to our method; it also affects other molecular diagnostic techniques, such as PCR-RFLP and real-time PCR, which face challenges in differentiating closely related species. Further studies involving a broader range of species are necessary to improve the reliability and practicality of these approaches for comprehensive species identification.

Of the 43 primer pairs designed in this study, 10 did not amplify any products (S2 Table). Several factors may have contributed to this outcome. One possible explanation is the presence of sequence mismatches at the primer binding sites, caused by genetic variation within or between populations, which may lead to inefficient or failed primer annealing. Additionally, secondary structures within the template DNA may hinder primer binding or polymerase extension. Another contributing factor may be suboptimal primer design parameters, such as melting temperature or GC content, despite careful optimization. To enhance the robustness of primer sets, their efficiency should be tested across a broader range of populations and closely related species.

In summary, we developed a bioinformatics method for extracting species-specific primers from transcriptome RNA-Seq data. Our results show that species-specific primers can be designed using RNA-Seq assemblies even for non-model organisms whose genomes have not been sequenced. These species-specific primers may be used for practical species discrimination using PCR. We identify two ways to improve our method. First, our method requires the manual design of primers to pair with species-specific primer candidate. A program that automatically extracts both primers would be more useful. Second, the method becomes cumbersome when distinguishing larger numbers of species, as each species requires individual PCR reactions for validation. Modifying the method for use with multiplex PCR would make it simpler to use.

## Supporting information

**S1 Table. Summary of sequence data, *de novo* assembly, and *k*-mer analysis.**
(XLSX)

**S2 Table. Species-specific primer candidates and results of PCR amplification.**
(XLSX)

**S1 File. Aligned sequences in FASTA format.**
(ZIP)

**S2 File. Species-specific primer candidates.**
(ZIP)

**S3 File. Electropherogram of agarose gel showing PCR amplifications from spider mites.** Lanes 1: *P. citri*, 2: *P. mori*, 3: *P. ulmi*, 4: *P. osmanthi*, 5: *S. shii*, 6: *E. nomurai*, 7: *E. celtis*, 8: *O. castaneae*, 9: *O. ilicis*, 10: *O. coffeae*, 11: *O. gotohi*, 12: *O. amiensis*, 13: *T. kanzawai*, 14: *T. parakanzawai*, 15: *T. urticae* (red form), 16: *T. urticae* (green form), 17: *T. truncatus*, 18: *T. puераricola*, 19: *T. piercei*.
(PDF)

**S1 Raw Images. All raw gel images in this study.**
(PDF)

## Acknowledgments

We thank Prof. Dr. N. Ogata (Nihon BioData Corporation, Kawasaki, Kanagawa, Japan) for proposing the idea of designing primers using *k*-mer analysis, and Dr. S. Ohno (Okinawa Prefectural Agricultural Research Center, Ishigaki, Okinawa, Japan) for collecting the spider mites.

## Author contributions

**Conceptualization:** Tomoko Matsuda.

**Data curation:** Tomoko Matsuda.

**Formal analysis:** Tomoko Matsuda.

**Funding acquisition:** Tetsuo Gotoh.

**Investigation:** Tomoko Matsuda, Hironori Sakamoto, Takumi Kayukawa, Yasuki Kitashima, Toshinori Kozaki.

**Methodology:** Tomoko Matsuda, Hironori Sakamoto, Takumi Kayukawa.

**Project administration:** Tomoko Matsuda, Tetsuo Gotoh.

**Resources:** Tomoko Matsuda, Hironori Sakamoto, Takumi Kayukawa, Yasuki Kitashima, Toshinori Kozaki, Tetsuo Gotoh.

**Software:** Tomoko Matsuda.

**Supervision:** Tetsuo Gotoh.

**Validation:** Tomoko Matsuda, Hironori Sakamoto, Takumi Kayukawa.

**Visualization:** Tomoko Matsuda.

**Writing – original draft:** Tomoko Matsuda.

**Writing – review & editing:** Tomoko Matsuda, Tetsuo Gotoh.

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
