## [Decision Letter · Decision Letter 0]

21 Nov 2024

PONE-D-24-26127A PCR primer design method for identifying spider mite species using the k-mer counting algorithmPLOS ONE

Dear Dr. Matsuda,

Thank you for submitting your manuscript to PLOS ONE. After careful consideration, we feel that it has merit but does not fully meet PLOS ONE’s publication criteria as it currently stands. Therefore, we invite you to submit a revised version of the manuscript that addresses the points raised during the review process.

Both reviewers have found your manuscript to have important merit, however, they also highlight important deficiencies. In particular, please refer to comments and suggestions by reviewer 2 to address questions related to study design,

We look forward to receiving your revised manuscript.

Kind regards,

Ulrike Gertrud Munderloh, Ph.D.

Academic Editor

PLOS ONE

Journal requirements:    When submitting your revision, we need you to address these additional requirements. 1. Please ensure that your manuscript meets PLOS ONE's style requirements, including those for file naming. The PLOS ONE style templates can be found at https://journals.plos.org/plosone/s/file?id=wjVg/PLOSOne_formatting_sample_main_body.pdf and https://journals.plos.org/plosone/s/file?id=ba62/PLOSOne_formatting_sample_title_authors_affiliations.pdf 2. PLOS ONE now requires that authors provide the original uncropped and unadjusted images underlying all blot or gel results reported in a submission’s figures or Supporting Information files. This policy and the journal’s other requirements for blot/gel reporting and figure preparation are described in detail at https://journals.plos.org/plosone/s/figures#loc-blot-and-gel-reporting-requirements and https://journals.plos.org/plosone/s/figures#loc-preparing-figures-from-image-files. When you submit your revised manuscript, please ensure that your figures adhere fully to these guidelines and provide the original underlying images for all blot or gel data reported in your submission. See the following link for instructions on providing the original image data: https://journals.plos.org/plosone/s/figures#loc-original-images-for-blots-and-gels.   In your cover letter, please note whether your blot/gel image data are in Supporting Information or posted at a public data repository, provide the repository URL if relevant, and provide specific details as to which raw blot/gel images, if any, are not available. Email us at plosone@plos.org if you have any questions. 3. Please remove your figures from within your manuscript file, leaving only the individual TIFF/EPS image files.  These will be automatically included in the reviewer’s PDF. (Figure 3 duplicated in Supportingfile_OriginalGel_Figure). 4. Please note that your Data Availability Statement is currently missing [the repository name and/or the DOI/accession number of each dataset OR a direct link to access each database]. If your manuscript is accepted for publication, you will be asked to provide these details on a very short timeline. We therefore suggest that you provide this information now, though we will not hold up the peer review process if you are unable. 

Reviewers' comments:

Reviewer's Responses to Questions

**Comments to the Author**

1. Is the manuscript technically sound, and do the data support the conclusions?

Reviewer #1: Yes

Reviewer #2: Yes

2. Has the statistical analysis been performed appropriately and rigorously? 

Reviewer #1: Yes

Reviewer #2: N/A

3. Have the authors made all data underlying the findings in their manuscript fully available?

Reviewer #1: Yes

Reviewer #2: Yes

4. Is the manuscript presented in an intelligible fashion and written in standard English?

Reviewer #1: Yes

Reviewer #2: Yes

5. Review Comments to the Author

Reviewer #1: The authors have developed a bioinformatics method for extracting species-specific primer candidate sequences from RNA Seq data sets of 19 spider mite species (Acari, Tetranychidae). By using the k-mer counting algorithm, the authors obtained between 257 and 48,621 species-specific unpaired primer candidates for the 19 spider mite species, and manually obtained a second primer that was also species-specific. The primer pairs were then onfirmed to work in the target species and to not work in the non-target species. This technique was effective in discriminating spider mite species and very useful in the lab for fast identification of spider mite species. The design and the statistical analysis was sound and approprite.

I have several suggestions for the paper.

1. The genomes of spider mites have been studied in more than four species. Please update the information in the Introduction section and cite more related articles.

2. Add more explanation about not The discussion section about the species-specific primer pairs were not obtained for two species of the 19 species tested.

I would suggest the acceptance of the article for publication in PLoS One after minor revision.

Reviewer #2: See attachment, because bullet points and formatting are not retained in copy paste.

6. PLOS authors have the option to publish the peer review history of their article (what does this mean? ). If published, this will include your full peer review and any attached files.

**Do you want your identity to be public for this peer review?** For information about this choice, including consent withdrawal, please see our Privacy Policy .

Reviewer #1: **Yes: ** Xiao-Yue Hong

Reviewer #2: **Yes: ** Marilou Boddé

---

## [Author Response · Author response to Decision Letter 1]

24 Jan 2025

In response to the journal’s requirements:

1. We have formatted the manuscript according to PLOS ONE’s style templates, including adjustments to the title and author sections. The document now adheres to the specified formatting guidelines, including Letter paper size, Times New Roman font (12 pt), and double-spaced lines.

2. We have reviewed the guidelines for preparing figures and blot/gel data. All raw gel images underlying our figures have been provided as Supporting Information (S1 Raw images). Additionally, we have detailed any unavailable raw image data for clarity.

3. Figures have been removed from the main manuscript and are submitted as individual TIFF files. The file previously named "Supportingfile_OriginalGel_Figure.pdf" has been revised to "S1 Raw images.pdf," with the duplication of Figure 3 removed.

4. The Sequence data used in the study have been updated with their DRA accession numbers and links.

Line 113: DRA007145 (https://www.ncbi.nlm.nih.gov/sra/?term=DRA007145)

Line 124: DRA018635 (https://www.ncbi.nlm.nih.gov/sra/?term=DRA018635)

Line 237: We are currently in the process of obtaining accession numbers for the sequences of PCR amplicons described in the Materials & Methods section. We can provide this information as soon as the accession numbers become available.

In response to Reviewers:

Comments 1

1. Consider changing the title to “A PCR primer design method for identifying spider mite species using k-mers” or “... using k-mer counts” or “... using k-mer counting”. There are several algorithms for k-mer counting and there is no discussion of those in the paper and that is also not necessary. What matters for the paper is the concept of unique k-mers, not the algorithm used to identify those.

Response 1

We revised the title accordingly. Please refer to line 4.

Before Revision

A PCR primer design method for identifying spider mite species using the k-mer counting algorithm

After Revision (line 4, Revised MS (clean copy))

A PCR primer design method for identifying spider mite species using the k-mer counting

Comments 2

2. Describe literature for species diagnostic PCR and the way it is used to diagnose species. There are instances where the PCR products have different known lengths for the different species and the diagnosis is based on comparing the query length to the expected lengths for each species. There are instances where the PCR product is sequenced and certain known variants are used to diagnose the species. In this case, the PCR primers should only work in a single species so whether or not PCR amplification is observed is used to diagnose species.

Response 2

The points you raised regarding the use of species- specific diagnostic PCR methods have been incorporated into the manuscript. Please refer to line 37.

Before Revision

Among its many uses, the polymerase chain reaction (PCR) can be used for distinguishing morphologically similar species. Designing the PCR primers to be specific for the target is essential.

After Revision (line 37, Revised MS (clean copy)))

Polymerase chain reaction (PCR) is a method for amplifying DNA fragments in vitro [1]. Because of its convenience and low cost, PCR has become a standard technique not only for biomedical, infectious disease, and forensic applications but also in other fields of biology, such as agricultural and ecological research. Among its many uses, PCR can be used to distinguish morphologically similar species. For example, PCR-RFLP (restriction fragment length polymorphism) distinguishes closely related species by using a single restriction enzyme to produce fragments of different lengths from DNA markers [2-4]. Another method is to design species-specific primers that can only be amplified in certain species and to identify the species based on successful PCR amplification and correct product size [5-7].

1. Mullis K, Faloona F. Specific synthesis of DNA in vitro via a polymerase-catalyzed chain reaction. Methods Enzymol. 1987;155: 335–350.

2. Clark TL, Meinke LJ, Foster JE. PCR–RFLP of the mitochondrial cytochrome oxidase (subunit I) gene provides diagnostic markers for selected Diabrotica species (Coleoptera: Chrysomelidae). Bull Entomol Res. 2001;91: 419–427.

3. Asraoui JF, Sayar NP, Knio KM, Smith CA. Fly diversity revealed by PCR-RFLP of mitochondrial DNA. Biochem Mol Biol Educ. 2008;36: 354–362.

4. Chua TH, Chong YV, Lim SH. Species determination of Malaysian Bactrocera pests using PCR-RFLP analyses (Diptera: Tephritidae). Pest Manag Sci. 2010;66: 379–384.

5. Lu W-N, Wu Y-T, Kuo M-H. Development of species-specific primers for the identification of aphids in Taiwan. Appl Entomol Zool (Jpn). 2008;43: 91–96.

6. Zhang T, Wang Y-J, Guo W, Luo D, Wu Y, Kučerová Z, et al. DNA barcoding, species-specific PCR and real-time PCR techniques for the identification of six Tribolium pests of stored products. Sci Rep. 2016;6: 28494.

7. Zhao Z-H, Cui B-Y, Li Z-H, Jiang F, Yang Q-Q, Kučerová Z, et al. The establishment of species-specific primers for the molecular identification of ten stored-product psocids based on ITS2 rDNA. Sci Rep. 2016;6: 21022.

Comments 3

3. Motivate why the primers are designed from RNA-Seq data and not from whole genome short read data. The authors briefly mention why they don’t work with genome assemblies (because these are available for fewer species than the RNA-Seq data), but they do not consider whole genome DNA data (without necessarily making this into a genome assembly, but rather work with the reads directly or aligned to an available genome assembly of a related species). Using whole genome data would probably result in more available targets because A) a lot more sequence is considered and B) non-transcribed regions are often less constrained in accumulating variation so likely there are many more species-informative regions among non-transcribed regions.

Response 3

We have added additional information about whole-genome short-read data in the Introduction section. Please refer to line 69.

Before Revision

On the other hand, transcriptome RNA-Seq data for more than 50 species in the family Tetranychidae are publicly available [22].

After Revision (line 69, Revised MS (clean copy))

Using whole genome data would likely increase the number of available targets and the number of sequences and non-transcribed regions considered. However, publicly available sequencing resources for spider mites (Acari, Tetranychidae) are limited, with only 13 species having whole genome short-read data deposited in the SRA, compared to 79 species with available RNA-Seq data [29] (according to the NCBI SRA database accessed on 24 December 2024). In a previous study, we performed RNA-Seq analysis on 72 species (73 strains) of spider mites. The resulting data have been made publicly available in DRA/SRA/ERA databases under the accession number DRA007145.

Comments 4-a

4. The methods are a bit difficult to follow. To my understanding, RNA reads are assembled and contigs would represent an observed transcript. Then, by orthology search from the annotated T. urticae genome, those contigs corresponding to an intronless gene are selected, and alignments of the 19 species are generated for each gene. On those selected contigs, 30-mers are counted and unique 30-mers were used for primer design. The paired primer was selected manually.

a. Why are alignments done separately for CDS and 5’ and 3’ sequence, also using different alignment methods?

Response 4-a

It is common practice to align coding sequences (CDSs) and untranslated regions (UTRs) separately, as demonstrated in the following previous studies. This approach is necessary because obtaining full-length transcripts across all species is challenging when using Illumina short-read sequencing technology.

Chua, B. H., McMinn, P. C., Lam, S. K. & Chua, K. B. Comparison of the complete nucleotide sequences of echovirus 7 strain UMMC and the prototype (Wallace) strain demonstrates significant genetic drift over time. J. Gen. Virol. 82, 2629–2639 (2001).

Wang, X.-W. et al. Transcriptome analysis and comparison reveal divergence between two invasive whitefly cryptic species. BMC Genomics 12, 458 (2011).

Cheng, J., Zhang, L., Hui, M., Li, Y. & Sha, Z. Insights into adaptive divergence of Japanese mantis shrimp Oratosquilla oratoria inferred from comparative analysis of full-length transcriptomes. Front. Mar. Sci. 9, 975686 (2022).

CDSs were aligned using DIALIGN-TX, which accounts for amino acid sequences to improve alignment accuracy. In contrast, 3' and 5' UTR were aligned using MAFFT, a widely used alignment tool, as considering amino acid sequences is not necessary for these non-coding regions. We have mentioned to these in the Materials & Methods section. Please refer to line 201.

Added text (line 201, Revised MS (clean copy))

CDSs were aligned using DIALIGN-TX, which considers amino acid sequences to improve alignment accuracy, as described in the section 'Identification of putative orthologous genes.' In contrast, 3' and 5' UTR sequences were aligned using MAFFT [56], a widely used alignment tool, as considering amino acid sequences is not necessary for these non-coding regions.

Comments 4-b

b. In figure 2 there seems to be a step missing, the phylogenetic trees are constructed to distinguish orthologous genes from paralogous genes, true? Also please mention what the distinguishing characteristic is between orthologous and paralogous genes in a phylogenetic tree (now it just says ‘identified visually’).

Response 4-b

Despite recent advances, all existing ortholog detection methods still suffer from false positive and false negative rates. For example, OrthoFinder is widely used to estimate orthologous genes, providing robust phylogenetic inference and orthogroup identification. However, by definition, orthogroups identified using this method may contain both orthologs and paralogs, as OrthoFinder groups all related sequences together without fully distinguishing between the two. Therefore, it is not always possible to completely exclude paralogs entirely using this approach.

We therefore constructed a phylogenetic tree to distinguish between orthologous and paralogous genes. A gene was classified as a paralog if its phylogenetic relationship clearly deviated from the established relationships within known spider mite family Tetranychidae

For example, as illustrated in the figure below, if the phylogenetic tree showed that the sequence from "T. truncatus (1765_T_tru_com938.seq3.p1)" does not cluster with sequences from other species in a manner consistent with their known evolutionary relationships, it was identified as a paralog rather than an ortholog. If a phylogenetic tree included paralogs, the genes used to construct that tree were excluded from the analysis. This visual assessment was based on the topology of the tree and the divergence of specific clades.

We have added these to the Materials & Methods section. Please refer to line 144.

Before Revision

Then, phylogenetic trees based on each orthologous gene were constructed using RAxML [45].

After Revision (line 144, Revised MS (clean copy))

Despite recent advances, existing ortholog detection methods still suffer from false-positive and false-negative rates. We therefore constructed a phylogenetic tree using RAxML [52] to differentiate between orthologous and paralogous genes. A gene was classified as a paralog if its phylogenetic relationship distinctly deviated from the established relationships among known spider mite species in the family Tetranychidae [29].

Comments 4-c

c. In the Extracting unique k-mers section the criteria could be ordered differently for ease of understanding.

i. Start with requirement 9, which is the crucial step, namely retain only unique k-mers (unique within and across species). NB: the ‘and’ should be replaced by ‘or’ to achieve (unique within and across species).

ii. Then introduce/motivate the remaining requirements: they are important for primer binding/open chromatin or other reasons why they were implemented.

iii. In requirement 6,7,8 replace ‘and’ by ‘or’. A number cannot be smaller than 11 and greater than 17 at the same time.

iv. Add ‘primer melting Temperature’ in brackets after Tm.

Response 4-c

We agree with your observations. As a result, we have revised the structure of these sections to follow a more logical order, ensuring a clearer presentation of each item and its purpose. Please refer to line 158.

Before Revision

The k-mers obtained from Jellyfish were filtered by the following requirements.

1. When a k-mer contained four or more runs of a single base (e.g., AAAAA and GGGG), the k-mer was removed.

2. When both 5' and 3' ends of a k-mer were A or T, the k-mer was removed.

3. When both 5' and 3' ends of a k-mer were three or more of G or C, the k-mer was removed.

4. When both 5' and 3' ends of a k-mer were three or more of A or T, the k-mer was removed.

5. When a k-mer contained more than three times of dinucleotide repeats (e.g., ATATAT and CGCGCG), the k-mer was removed.

6. When the GC content of full-length k-mer (30 mers) was less than 11 and greater than 17, the k-mer was removed.

7. When the GC content of the former and latter half of k-mer (15 mers) was less than four and greater than 10, the k-mer was removed.

8. When the Tm value of k-mer was less than 67 and greater than 82, the k-mer was removed.

9. When a k-mer matched another k-mer within the species and a k-mer of another species, the k-mer was removed.

After Revision (line 158, Revised MS (clean copy))

Primer specificity for species-specific detection:

1. If a k-mer matched another k-mer within the same species or a k-mer of another species, the k-mer was removed.

Tm (primer melting temperature):

2. If the Tm value of a k-mer was less than 67 or greater than 82, the k-mer was removed.

GC Content: Preferably in the range of 40%–60%. Include 1–2 G or C bases at the 3' end to improve binding stability; however, avoid excessive GC content at the 3' terminus to prevent non-specific binding.

3. If the GC content of a full-length k-mer (30-mers) was less than 11 or greater than 17, the k-mer was removed.

4. If the GC content of the former or latter half of a k-mer (15-mers) was less than four or greater than 10, the k-mer was removed.

5. If both the 5' and 3' ends of a k-mer were A or T, the k-mer was removed.

6. If both the 5' and 3' ends of a k-mer contained three or more of G or C, the k-mer was removed.

7. If both the 5' and 3' ends of a k-mer contained three or more of A or T, the k-mer was removed.

Avoid Repeats and Runs: Prevent intra-primer and inter-primer complementarity to reduce primer-dimer formation.

8. If a k-mer contained four or more runs of a single base (e.g., AAAAA or GGGG), the k-mer was removed.

9. If a k-mer contained more than three times of dinucleotide repeats (e.g., ATATAT or CGCGCG), the k-mer was removed.

Comments 4-d

d. In the Filtering k-mers section, it seems a bit roundabout to use bowtie and blastn to filter k-mers for similarity to the contigs. Under the hood this can probably also be phrased as a k-mer matching criterion. I’m curious to know whether you really need bowtie and blast to get a well filtered set or whether by tweaking the parameters you can get the same level of filtering by just using one of these programs.

R

---

## [Decision Letter · Decision Letter 1]

7 Feb 2025

PONE-D-24-26127R1A PCR primer design method for identifying spider mite species using the k-mer countingPLOS ONE

Dear Dr. Matsuda,

Thank you for submitting your manuscript to PLOS ONE. After careful consideration, we feel that it has merit but does not fully meet PLOS ONE’s publication criteria as it currently stands. Therefore, we invite you to submit a revised version of the manuscript that addresses the points raised during the review process.

There are a few issues that need clarification, please see reviewer 2 comments. If possible, please incorporate your response into appropriate sections of the manuscript. 

We look forward to receiving your revised manuscript.

Kind regards,

Ulrike Gertrud Munderloh, Ph.D.

Academic Editor

PLOS ONE

Journal Requirements:

Reviewers' comments:

Reviewer's Responses to Questions

**Comments to the Author**

1. If the authors have adequately addressed your comments raised in a previous round of review and you feel that this manuscript is now acceptable for publication, you may indicate that here to bypass the “Comments to the Author” section, enter your conflict of interest statement in the “Confidential to Editor” section, and submit your "Accept" recommendation.

Reviewer #1: All comments have been addressed

Reviewer #2: (No Response)

2. Is the manuscript technically sound, and do the data support the conclusions?

Reviewer #1: Yes

Reviewer #2: Yes

3. Has the statistical analysis been performed appropriately and rigorously? 

Reviewer #1: Yes

Reviewer #2: N/A

4. Have the authors made all data underlying the findings in their manuscript fully available?

Reviewer #1: Yes

Reviewer #2: Yes

5. Is the manuscript presented in an intelligible fashion and written in standard English?

Reviewer #1: Yes

Reviewer #2: Yes

6. Review Comments to the Author

Reviewer #1: Thank you for answering all my questions and explaing the comments. I agree with your revision and would suggest to accept the revised manuscript for publication in PLoS One.

Reviewer #2: I would remove "the" from the title, so:

A PCR primer design method for identifying spider mite species using the k-mer counting

Re comment 5: did you recycle any of the methods from the initial candidate primer identification steps to check the specificity/uniqueness of the complementary primer?

Re comment 6: are there any heuristics for which primers pairs to pick first or was that completely at random? And then select another pair at random if the first pair failed?

All other comments have been satisfactorily addressed.

7. PLOS authors have the option to publish the peer review history of their article (what does this mean? ). If published, this will include your full peer review and any attached files.

**Do you want your identity to be public for this peer review?** For information about this choice, including consent withdrawal, please see our Privacy Policy .

Reviewer #1: **Yes: ** Xiao-Yue Hong

Reviewer #2: No

---

## [Author Response · Author response to Decision Letter 2]

16 Feb 2025

In response to the journal’s requirements:

We have thoroughly reviewed our reference list to ensure its completeness and accuracy, and we confirm that no retracted articles are cited. Additionally, we have rechecked the formatting and made minor adjustments as necessary.

Furthermore, we have made the following revisions:

・Added clarifications to the manuscript in response to the reviewers' comments.

・Registered the detailed experimental procedures on protocols.io. Once the review process on protocols.io is complete, we will include the DOI in the manuscript.

・Added a column in Table 2 for entering accession numbers. We are currently in the process of obtaining these numbers and are awaiting a response from the DNA Data Bank of Japan (DDBJ).

・Identified and corrected duplicate entries in Table S2. Consequently, we have updated the manuscript to reflect the correct number of primers, revising the description from "45 primers" to "43 primers."

In response to Reviewers:

Reviewer #2's comments 1

I would remove "the" from the title, so:

A PCR primer design method for identifying spider mite species using the k-mer counting

Response 1

In the manuscript, we had already removed 'the'; however, the title entered in the submission form still included it. The manuscript already contained the correct title, so we have now updated the title in the submission form accordingly.

Reviewer #2's comments 2

Re comment 5: did you recycle any of the methods from the initial candidate primer identification steps to check the specificity/uniqueness of the complementary primer?

Response 2

We appreciate your suggestion and have conducted an additional BLASTN search to further verify the specificity and uniqueness of the complementary primer. This step has been incorporated into the manuscript.

Added text (line 210, Revised MS (clean copy))

To confirm the uniqueness of the species-specific primers, we conducted additional BLASTN searches to compare the species-specific primers with the contigs from the 19 assemblies after clustering with CD-HIT.

Reviewer #2's comments 3

Re comment 6: are there any heuristics for which primers pairs to pick first or was that completely at random? And then select another pair at random if the first pair failed?

All other comments have been satisfactorily addressed.

Response 3

The primer pairs were primarily designed at random, including cases where the initial primer failed. However, since alignments were checked when designing the primer pairs, if an alignment was particularly suitable for primer design, additional primers were sometimes designed from the same contig. As a result, there are cases where primers were designed from the same transcript, such as XM_015937235.2 (tetur01g00100) and NW_015449938.1 (tetur09g06570) in S2 Table. We have also incorporated this clarification into the manuscript.

Added text (line 259, Revised MS (clean copy))

The primer pairs were initially designed randomly. However, since sequence alignments were carefully examined during the primer design process, additional primers were occasionally designed from the same contig when a particular alignment was deemed highly suitable for primer design. As a result, multiple primers were designed from the same transcript in certain cases, such as XM_015937235.2 and NW_015449938.1, as shown in Table 2 and S2 Table. Finally, 43 PCR primers were designed to target species-specific sequences.

---

## [Editor Report · Decision Letter 2]

3 Mar 2025

A PCR primer design method for identifying spider mite species using k-mer counting

PONE-D-24-26127R2

Dear Dr. Matsuda,

We’re pleased to inform you that your manuscript has been judged scientifically suitable for publication and will be formally accepted for publication once it meets all outstanding technical requirements.

Kind regards,

Ulrike Gertrud Munderloh, Ph.D.

Academic Editor

PLOS ONE
---

## [Editor Report · Acceptance letter]

PONE-D-24-26127R2

PLOS ONE

Dear Dr. Matsuda,

I'm pleased to inform you that your manuscript has been deemed suitable for publication in PLOS ONE. Congratulations! Your manuscript is now being handed over to our production team.

Kind regards,

on behalf of

Dr. Ulrike Gertrud Munderloh

Academic Editor

PLOS ONE